# Novel Evolutionary Engineering Approach to Alter Substrate Specificity of Disaccharide Transporter Mal11 in *Saccharomyces cerevisiae*

**DOI:** 10.3390/jof8040358

**Published:** 2022-03-30

**Authors:** Sophie Claire de Valk, Robert Mans

**Affiliations:** Department of Biotechnology, Delft University of Technology, Van der Maasweg 9, 2629 HZ Delft, The Netherlands; s.c.devalk@tudelft.nl

**Keywords:** sugar transport, counter-selection, proton symport, mutation, transport protein

## Abstract

A major challenge in the research of transport proteins is to understand how single amino acid residues contribute to their structure and biological function. Amino acid substitutions that result in a selective advantage in adaptive laboratory evolution experiments can provide valuable hints at their role in transport proteins. In this study, we applied an evolutionary engineering strategy to alter the substrate specificity of the proton-coupled disaccharide transporter Mal11 in *Saccharomyces cerevisiae*, which has affinity for sucrose, maltose and glucose. The introduction of *MAL11* in a strain devoid of all other sugar transporters and disaccharide hydrolases restored growth on glucose but rendered the strain highly sensitive to the presence of sucrose or maltose. Evolution in glucose-limited continuous cultures with pulse-wise addition of a concentrated sucrose solution at increasing frequency resulted in the enrichment of spontaneous mutant cells that were less sensitive to the presence of sucrose and maltose. Sequence analysis showed that in each of the two independent experiments, three mutations occurred in *MAL11*, which were found responsible for the disaccharide-insensitive phenotype via reverse engineering. Our work demonstrates how laboratory evolution with proton-motive force-driven uptake of a non-metabolizable substrate can be a powerful tool to provide novel insights into the role of specific amino acid residues in the transport function of Mal11.

## 1. Introduction

Over the course of evolution, a myriad of sugar transporters have emerged with diverse mechanisms, kinetics, regulation and substrate specificities [1,2]. Whereas some transporters are highly specific towards one sugar substrate, such as the fructose-specific Ffz1 from *Zygosaccharomyces rouxii* [3], others are able to mediate the uptake of multiple sugars, such as Mal11 from *Saccharomyces cerevisiae* [4,5,6]. Knowledge on the three-dimensional organization of amino acid residues and how they interact with each other to bind and translocate substrates is highly relevant to understand how transporters with broad substrate range evolved into specific transporters (or vice versa) in nature. Since sugars are important substrates of industrial biotechnology, their transporters are also relevant targets in engineering strategies for the improvement of microbial cell factories [7,8,9,10]. Ideally, transport proteins (and thus their encoding genes) can be designed at will, to obtain strains with unrestricted transport of the desired substrate without undesired inhibition or competition by other molecules. This scenario requires extensive knowledge on the structure and mechanism of transport proteins and the function of individual amino acid residues on their overall activity.

The steadily increasing number of resolved protein structures through methods such as X-ray crystallography has contributed to our understanding of the tertiary structure of membrane proteins [11,12]. Still, compared to soluble proteins, membrane protein structures are rather under-represented in the protein data bank, partially due to the difficulty of membrane protein crystallization [13]. In addition, computational methods for protein structure prediction, based on the crystal structure of a homologous protein, have provided valuable insights [14], and promising results are now obtained using the AI system AlphaFold [15]. Knowledge on the role of individual amino acid residues can be acquired by studying mutant transporter variants acquired via (targeted) mutagenesis [16,17] or adaptive laboratory evolution (ALE) [18,19,20]. For the latter approach, cultivation conditions are carefully designed to allow for the enrichment of mutants with a desired phenotype. By analyzing the genome of evolved strains, mutations can be identified that underlie the evolved phenotype, and (point) mutations in transport proteins hint at an important role of specific amino acid residues for transport function. This method is especially powerful when the investigated transport process is essential for growth [16,20]. For example, ALE has been applied to improve kinetics of proteins that mediate the uptake of an essential substrate by selecting for mutants with an improved growth rate through serial propagation in batch conditions [21,22,23,24]. Alternatively, transporters with improved affinity can be obtained by prolonged cultivation in substrate-limited continuous cultures [25,26]. Some evolutionary engineering approaches were aimed at specializing transporters with broad substrate affinity for one of its substrates. For example, Hxt transporters in *S. cerevisiae*, which transport glucose and xylose (albeit with moderate affinity and glucose inhibition), were specialized for xylose transport by evolving a hexokinase-deficient strain for growth on xylose in the presence of glucose [27,28,29]. The resulting mutant transporter variants were found to contain amino acid substitutions that decreased or abolished the affinity of the transporter for glucose but not for xylose. However, complete abolishment of glucose affinity was accompanied by a ~60% reduction of the V_max_ for xylose.

A more stringent selection for specialized transporters could be achieved if transport of the undesired substrate can be made toxic to the cells (counter-selection), while transport of the desired substrate provides an evolutionary benefit. Transport of a metabolizable sugar is usually not toxic, but exceptions can be found when sugar uptake is coupled to the uptake of a proton (proton symport). Uptake via such a proton-symporter is not only driven by the concentration gradient of the sugar, but also by the chemical and electrical potential difference of the proton-motive force (pmf). At typical values of the pmf in *S. cerevisiae* (−150 to −200 mV [30]), this driving force could enable a 1000-fold accumulation of sugar inside the cell. Such accumulation would lead to osmotic bursting of cells, and therefore excessive substrate accumulation is prevented by regulation of transporters and enzymes that catalyze the intracellular degradation of the sugar. The importance of this regulation is demonstrated by the phenomenon referred to as ‘substrate-accelerated cell death’, which has for example been observed when a pulse of maltose was administered to prolonged, maltose-limited *S. cerevisiae* chemostat cultures [26,31]. The sudden switch from maltose-limited to maltose-excess conditions decreased culture viability by up to 70%, which was hypothesized to be due to uncontrolled, sudden and fast uptake of maltose and protons into the cells, leading to rapid acidification and an osmotic burst.

In this study, we investigated whether the proton-motive force could be employed to create a selective pressure against substrate uptake by proton-coupled transporters. To this end, we introduced *MAL11*, which encodes a proton-symporter with affinity for sucrose, maltose and glucose [4,5,6] as the sole sugar transporter gene in an *S. cerevisiae* strain that is not able to hydrolyze sucrose or maltose. Since these disaccharides are not metabolized, we expected that cells would suffer negative effects from their intracellular accumulation, which is ‘forced’ by the proton-motive force due to proton-coupled uptake. By evolving this strain in glucose-limited chemostats with increasing addition of sucrose, we selected for mutants that were less sensitive to the presence of sucrose, while maintaining their ability to take up glucose. This evolution was followed by an analysis of mutations and growth characterization of evolved and reverse engineered strains in medium with glucose, supplemented with various concentrations of sucrose or maltose.

## 2. Materials and Methods

### 2.1. Strains and Maintenance

All *Saccharomyces cerevisiae* strains used in this study are derived from the CEN.PK lineage [32]. For long-term storage, glycerol was added to cells that were grown until late exponential phase, to obtain a final concentration of 30% (*v*/*v*) glycerol, after which 1 mL aliquots were stored at −80 °C. Plasmids were propagated in *Escherichia coli* XL1-Blue cells (Agilent, Santa Clara, CA, USA), which were also stored at −80 °C after addition of glycerol to a final concentration of 25% (*v*/*v*) to overnight cultures.

### 2.2. Molecular Biology Techniques

Plasmids were isolated from yeast strains using the Zymoprep Yeast Plasmid Miniprep II kit (Baseclear, Leiden, The Netherlands), after which 1 µL of this mixture was transformed, according to the manufacturer’s protocol, into XL1-Blue chemically competent *E. coli* cells (Agilent) for plasmid propagation. Plasmids were isolated from *E. coli* cells using the GeneJET Plasmid Miniprep Kit (Thermo Fisher Scientific, Waltham, MA, USA). Transformation of *S. cerevisiae* strains was performed using LiAc/ssDNA/PEG, as previously described [33]. After transformation, single colonies were re-streaked three consecutive times to ensure an isogenic single cell line. For whole-genome sequencing, genomic DNA was isolated from stationary *S. cerevisiae* cultures using the QIAGEN Blood & Cell Culture DNA Kit with 100/G Genomics-tips (Qiagen, Hilden, Germany) according to the manufacturer’s protocol. Whole-genome sequencing of these DNA samples was performed by Genomescan (Leiden, The Netherlands). Then, 350 bp insert libraries were constructed, which were paired-end sequenced (150 bp reads) with an Illumina HiSeq X sequencer. The data analysis was performed as described previously [34]. Sanger sequencing of genes was performed using the Macrogen (Amsterdam, the Netherlands) EZ-seq sequencing service according to the provided instructions.

### 2.3. Strain Construction

All *S. cerevisiae* strains used in this study are listed in Table 1.

Plasmid pUDE432 [36] was introduced into strain IMK1010, resulting in strain IMZ786, which was subsequently evolved in duplicate using independent bioreactor cultures. After evolution, the evolved population from each bioreactor was stocked, resulting in IMS1225 and IMS1226. Both strains were plated on SMD with 5 g L^−1^ sucrose and re-streaked three consecutive times. The resulting single-colony isolates were grown in liquid SMD with 5 g L^−1^ sucrose and stocked as IMS1230 and IMS1231, respectively. The plasmids pUDE1222 and pUDE1223 were isolated from IMS1230 and IMS1231, respectively, and reintroduced in IMK1010, resulting in strains IME753 and IME754, respectively.

### 2.4. Media and Cultivation

*E. coli* cultures were grown at 37 °C in LB medium, supplemented with 100 μg mL^−1^ ampicillin for selection and maintenance of plasmids. Yeast strains were grown on synthetic medium (SM), which was heat sterilized for 20 min at 121 °C, after which a filter-sterilized vitamin solution and 20 g L^−1^ glucose was added (SMD) [37]. Medium for anaerobic cultivations was additionally supplemented with 10 mg L^−1^ ergosterol and 420 mg L^−1^ Tween 80, which were added in from a concentrated solution (800×) in absolute ethanol [38]. For preparation of solid medium plates, 2% (*w*/*v*) agar was added to the media prior to heat sterilization. Medium used in bioreactor cultivations was additionally supplemented with 0.2 g L^−1^ Antifoam C (Sigma Aldrich, Saint Louis, MO, USA). Aerobic shake flask cultures were grown in 500 mL round bottom flasks with 100 mL medium, which were incubated in an Innova orbital shaker (Eppendorf, Nijmegen, the Nederlands) at 200 rpm at 30 °C. Laboratory evolution of IMZ786 was conducted in 2 L laboratory bioreactors (Applikon, Delft, the Netherlands) with a 1 L working volume. Cultures were stirred at 800 rpm, the temperature was controlled at 30 °C and the pH was kept constant at 5.0 through automated addition of 2.0 M KOH. The cultures were sparged with 500 mL N_2_ min^−1^ (<5 ppm O_2_), and medium vessels were sparged with nitrogen as well. To enable evolution in continuous culture set-up, SMD medium pumps were switched on after glucose depletion in a preceding batch phase, to obtain a constant flowrate. The volume was kept constant at 1 L using an effluent pump that was controlled by an electric level sensor, resulting in a stable dilution rate. After a stable CO_2_ concentration in the reactor off-gas was observed, a filter-sterilized 500 g L^−1^ sucrose solution was added to the culture in one-second pulses via a separate pump. Based on observations on the online measurements of the CO_2_ concentration in the reactor off-gas, the frequency of these pulses was manually increased and decreased to exert a selective pressure on the culture but prevent culture washout.

Maximum specific growth rates were determined in a Growth-Profiler system (EnzyScreen, Heemstede, The Netherlands) equipped with 96-well plates in a culture volume of 250 μL, set at 250 rpm and 30 °C. The measurement interval was set at 20 min. Raw green values were corrected for well-to-well variation using measurements of a 96-well plate containing a culture with an externally determined optical density, measured at 660 nm using a Libra S11 spectrophotometer (Biochrom, Cambridge, United Kingdom), of 4.19 in all wells. Optical densities were calculated by converting green values (corrected for well-to-well variation) using a calibration curve that was determined by fitting a third-degree polynomial through 71 measurements of cultures with known OD values between 0.75 and 24.5 (Appendix A). Growth rates were determined using the calculated optical densities of at least 10 points (corresponding to at least 200 min) in the exponential phase. Exponential growth was assumed when an exponential curve could be fitted with an R^2^ of at least 0.985. To validate the experimental procedure and data analysis, the growth rates of laboratory strain CEN.PK113-7D were also determined in all conditions, which corresponded to within 5% of previously determined values for this strain grown aerobically in SMD [39].

## 3. Results

### 3.1. Evolution for Decreased Sucrose Sensitivity

The previously constructed strain IMK1010 is devoid of all hexose transporters, disaccharide transporters and disaccharide hydrolases and is therefore unable to grow on any sugar substrate [35]. *MAL11*, encoding a proton-symporter with affinity for maltose, sucrose and glucose, was introduced in IMK1010 via the episomal expression vector pUDE432. Subsequently, growth of the resulting strain (IMZ786) was investigated on synthetic medium (SM) with only glucose, maltose or sucrose as the carbon source, and on SM with mixtures of glucose with either sucrose or maltose. Although introduction of *MAL11* was expected to enable the uptake of glucose, sucrose and maltose, in IMZ786 it only complemented the growth on medium with glucose (SMD, Figure 1). We hypothesized that the inability to grow on disaccharides was caused by the absence of disaccharide hydrolysis activity in IMZ786, which is essential for their further metabolism [40] (Appendix A). Moreover, IMZ786 was found to be sensitive to the presence of either sucrose or maltose, as could be observed from its inability to grow on SMD with the addition of 20 g L^−1^ of sucrose or maltose (Figure 1). We attributed this detrimental effect to the proton symport mechanism of the constitutively expressed Mal11, which enabled proton-motive force-driven intracellular accumulation of the disaccharides that, in this specific strain background, could not be relieved by their degradation or export. To investigate whether this inability to grow on SMD in the presence of sucrose or maltose was not only due to competitive inhibition of the transporter, cultures of IMK1010 (*sugar^0^*) and IMZ786 (*sugar^0^ MAL11*), which were pre-grown in SM with 2% (*v*/*v*) ethanol as the carbon source, were transferred to SM with 2% (*v*/*v*) ethanol and 20 g L^−1^ sucrose. While the optical density of IMK1010 in the presence of sucrose increased from 0.11 to 17.9 in 45.5 h, that of IMZ786 only increased from 0.07 to 0.12 within the same time span. These results indicate that growth on carbon sources not transported by Mal11, such as ethanol, which passively diffuses across the plasma membrane, is strongly impaired and likely caused by Mal11-mediated intracellular accumulation of sucrose. 

The sensitivity of IMZ786 towards sucrose when grown on glucose inspired us to investigate whether this property could enable a laboratory evolution strategy to decrease the substrate specificity of Mal11 for sucrose, while retaining its glucose transport activity. To this end, IMZ786 was first grown in two independent anaerobic glucose-limited continuous cultures at a dilution rate of 0.07 h^−1^, and online measurements of the CO_2_ concentrations in the reactor off-gas were used as a means to monitor metabolic activity and growth (Figure 2). After a stable off-gas CO_2_ concentration of ~0.83% was observed, we then investigated whether the presence of sucrose was also toxic to IMZ786 under glucose-limited conditions by separately adding a concentrated sucrose solution (500 g L^−1^) to the culture in one-second pulses, which resulted an average flowrate of ~16 mL h^−1^. Indeed, a near instant decrease of the CO_2_ production towards 0.08% was observed (within 1 h), indicating that growth and metabolic activity had ceased due to the addition of sucrose (Figure 2A). To prevent culture washout, the addition of sucrose was stopped until a stable CO_2_ concentration of ~0.85% was observed again. Then, the frequency at which the pulses of the sucrose solution were added was adjusted to obtain an average flowrate of ~0.09 mL h^−1^, which again led to a gradual decrease in CO_2_ production (Figure 2B), indicating significant sucrose toxicity in the culture. After approximately 480 h, the off-gas CO_2_ concentration first stabilized at 0.22%, and then increased to reach a new stable value at ~0.7% (Figure 2C). We hypothesized that this decrease, followed by an increase in CO_2_ production, was the result of washout of the original strain IMZ786, followed by the emergence and enrichment of less sensitive mutants. To increase the selective pressure on sucrose insensitivity, the frequency of the sucrose pulses was then manually increased (Figure 2D), after which the CO_2_ concentration was allowed to stabilize again (Figure 2E). This cycle was repeated until a stable CO_2_ output (0.53% for the first replicate, 0.68% for the second) was observed at an average flowrate of 1.6 mL h^−1^ (Appendix A), which, under the assumptions of steady-state, ideal mixing and that sucrose is not consumed, corresponded to approximately 11.4 g L^−1^ of sucrose in the reactor. After a total of approximately 350 generations, evolved populations were stocked from each reactor, resulting in IMS1225 and IMS1226, from which single cell lines were isolated (IMS1230 and IMS1231, respectively).

### 3.2. Evolved Strains Are Less Sensitive to Sucrose and Maltose

To investigate whether the bioreactor-derived strains were less sensitive to the presence of the disaccharides sucrose and maltose, the maximum specific growth rates of the evolved populations IMS1230 and IMS1231 were determined in SMD medium with various concentrations of either sucrose or maltose and compared to those of the unevolved parental strain IMZ786. Once again, the toxic effect that sucrose has on IMZ786 was apparent by the absence of growth on SMD in the presence of ≥2.5 g L^−1^ of sucrose, and only at a concentration of 0.16 g L^−1^ of sucrose no effect on the growth rate could be observed compared to growth on SMD without sucrose (Figure 1A).

On the contrary, the evolved strains IMS1225 and IMS1226 exhibited growth in all tested conditions, and in each condition, their growth rates were higher than that of the unevolved parental strain. For IMS1225, no effect on the growth rate could be observed between the SMD control and cultures with sucrose concentrations up to 0.625 g L^−1^, and for IMS1226 up to 1.25 g L^−1^ (Figure 1A). The presence of maltose had a similar effect on the growth rates of the three strains, although the toxicity effect was slightly less apparent than for sucrose (Figure 1B). For instance, the parental strain IMZ786 also exhibited growth in the presence of up to 2.5 g L^−1^ of maltose and compared to the growth rates of IMS1225 and IMS1226 on SMD, no effects were observed upon addition of up to 5 g L^−1^ and 10 g L^−1^ of maltose, respectively. Strikingly, the addition of 20 g L^−1^ of maltose only led to a 7% decrease of the growth rate of IMS1226. These results indicate that the evolution not only affected the sensitivity towards sucrose, but also its isomer maltose. Moreover, the growth rates of both strains on SMD without sucrose or maltose were 19.4% (IMS1225) and 10.1% (IMS1226) higher than that of the parental strain. In all conditions, growth rates of IMS1230 were highly similar to those of IMS1225, whereas the growth rates of IMS1231 were highly similar to those of IMS1226, indicating that the isolated single colonies are phenotypical representatives of the evolved population.

### 3.3. Mutations Occurred in MAL11 during Evolution

To investigate whether mutations occurred in the transporter gene *MAL11* during the laboratory evolution experiment, the plasmids from single-colony isolates IMS1230 and IMS1231 were isolated and the *MAL11* open reading frame was sequenced. Strikingly, three non-synonymous mutations had occurred in each independently-evolved strain: C490T, G682C and G1163C in IMS1230 and G682C, G1079C and G1163T in IMS1231 (Figure 3A). Although G1163 was mutated to a different base in each strain, the resulting amino acid substitution in the protein was identical, indicating that there were two common amino acid changes (Ala-228-Pro and Gly-388-Ala) in Mal11 that evolved independently in the two cultures. 

The locations of these mutations were investigated using a previously constructed homology model of Mal11 [41] (Figure 3B,C). The mutations that where shared in the two evolved strains were both found to be located in the central cavity of the protein, as well as the Leu-164-Phe mutation that was identified in IMS1230, albeit at different heights in the protein structure. To investigate the proximity of these mutations to the sugar binding sites, the Mal11 structure was compared to that of the homologous human GLUT3 sugar transporter, for which crystal structures have been determined for the glucose- and maltose-bound proteins [42]. These crystal structures showed that binding of one of the glucose units comprising maltose in GLUT3 completely overlaps with binding of a glucose molecule. Superimposition of the Mal11 homology model with the two GLUT3 structures suggested that the three mutations in the central cavity of Mal11 might be relatively close to the amino acids that form the sugar binding pocket (Figure 4). Strikingly, L164F and G388A appeared to be in near proximity to the glucose monomer of maltose that does not overlap with glucose.

### 3.4. Mutations in MAL11 Are Responsible for Decreased Sucrose and Maltose Sensitivity

To investigate whether the mutations that were identified in *MAL11* were responsible for the decreased sensitivity of the evolved strains towards sucrose and maltose, the plasmids containing the mutated variants of *MAL11* were isolated from IMS1230 and IMS1231 and reintroduced in the unevolved IMK1010 (*sugar^0^*) strain background. When the resulting strains (IME753 and IME754) were grown on SMD in the presence of different concentrations of sucrose or maltose (Figure 1), they performed similarly to their corresponding evolved strain from which the respective *MAL11* allele originated. These observations indicate that the mutations in *MAL11* are solely responsible for the evolved phenotype and no other genomic mutations contributed to the decreased sensitivity of evolved strains towards sucrose and maltose.

## 4. Discussion

In this study, we successfully applied an evolutionary engineering strategy to decrease the sensitivity of a *MAL11*-expressing strain lacking maltose or sucrose hydrolases towards sucrose while maintaining its ability to consume glucose. Concurrently, the sensitivity towards maltose was also decreased, which suggests that the mechanism behind the decreased sensitivity towards both disaccharides is highly similar. Expression of the evolved *MAL11* alleles in an unevolved strain background and characterization of the resulting strains confirmed that the mutations in *MAL11*, which result in amino acid changes in the Mal11 protein, underlie the evolved phenotype. We hypothesize that the improved growth performance in the presence of these sugars could be due to (one of) the following alterations: (1) a decreased binding affinity of Mal11 for disaccharides; (2) an overall lower disaccharide transport capacity of the protein, resulting in less toxic accumulation of disaccharides in growing cultures; (3) glucose-specialized uptake, due to glucose inhibition of disaccharide uptake, similar to what is observed for Hxt transporters in mixtures of glucose and pentose sugars in *S. cerevisiae* [7]; (4) loss of proton coupling to sugar uptake, resulting in a facilitated diffusion mechanism. In the latter case, intracellular accumulation of disaccharides is limited to their extracellular concentrations. Future work to investigate which mechanism is responsible for the evolved phenotype could involve the determination of kinetic parameters of the Mal11 variants (K_m_, K_i_ and V_max_) via transport assays with radiolabeled sucrose, maltose and glucose [41,43,44]. The accumulation ratio (ratio between intra- and extracellular sugar concentrations) combined with proton-uptake assays could also provide insight into the transport mechanisms of the evolved Mal11 proteins. Additionally, the effect of expressing these transporters on strain physiology, and more specifically, anaerobic biomass yield on sugar, can also be used to elucidate their transport mechanism [23,36,45].

Two identical amino acid substitutions (A228P and G388A) occurred in both independent evolution lines, which suggests that these two amino acid residues play an important role in determining the substrate specificity of Mal11 towards glucose and disaccharides and that their substitution resulted in the most substantial fitness benefit under the conditions of the evolution experiment. In previous work, it was found that in *Escherichia coli* XylE and the human GLUT transporters, which are homologous to Mal11, the position that corresponds to Gly-388 in Mal11 is occupied by a tyrosine residue [46]. In those transporters, this tyrosine residue is involved in closing off the sugar-binding site from the aqueous extracellular space [47,48]. Since in Mal11, a tyrosine residue is located one helix-turn above this position, the location of this hydrophobic residue may sterically dictate the size of the sugar that can be transported by the protein [46]. We speculate that the substitution of Gly-388 by alanine, which has a larger hydrophobic side chain, results in a functional similarity to the tyrosine residues occupying this position in the homologous monosaccharide transporters. A similar mechanism could be envisioned as a result of the L164F mutation, which was also predicted to be in close proximity to the ‘additional’ glucose unit of Mal11-bound maltose and does not overlap with glucose binding (Figure 4). Here, substitution of the leucine side chain by the bulky aromatic side chain of phenylalanine could sterically hinder the binding of disaccharides. The roles of the other amino acid residues in Mal11 that were mutated during evolution have not been predicted previously, which further demonstrates the power of evolutionary engineering to pinpoint amino acid residues with important roles in the transport mechanism. Testing the effects of the individual and different combinations of the mutations in Mal11 identified in this study on sucrose, maltose and glucose transport capacity could provide valuable insights into the importance of each mutation for the evolved phenotype and potential synergistic effects of mutation combinations.

To the best of our knowledge, this is the first demonstration of an evolutionary engineering strategy where proton-coupled substrate uptake is used as a counter-selective pressure to change the substrate specificity of a transporter. One of the mutations that we identified corresponded to an amino acid residue whose role could be predicted based on known crystal structures. This strategy could therefore also be valuable to identify important amino acid residues in other, less well studied transport proteins for which a crystal structure is not available yet. When selecting targets for such an evolutionary strategy, we consider three criteria to be especially important.

First, the transporter of interest should remain essential for growth under the selective conditions. Otherwise, cells are likely to ‘escape’ the selective pressure by evolving loss-of-function mutations in the targeted transporter. For example, when the aim of an evolution experiment is to evolve the amino acid permease Put4, which actively transports alanine, glycine and proline [49], into a proline-specific transporter, Put4-mediated proline transport can be made essential for growth by using a strain background in which all other related transporter genes have been deleted, as well as *PRO3*. The *PRO3* deletion results in a proline-auxotrophic strain that relies on uptake of proline from the medium [50], while transport of alanine and glycine, for which *S. cerevisiae* is prototrophic, can be lost.

Second, the counter-selectable ‘substrate’ should not be metabolized once taken up by cells, to allow for toxic accumulation. In this study, the absence of disaccharide hydrolases was sufficient to prevent intracellular conversion of sucrose or maltose. Another relatively straightforward application could be similar to the aforementioned transporters with affinity for both pentose and hexose sugars, where deletion of hexokinase-encoding genes is sufficient to prevent the conversion of glucose, while the introduction of a novel metabolic route allows for pentose conversion [27,28,29,51]. Such a strain design could therefore also be used for the evolution of proton-coupled hexose/pentose transporters, such as Gxs1 from *Candida intermedia* [52]. However, this criterium might be less straightforward with other substrates, especially when considering that the ability to grow on the essential substrate should be maintained. For example, when aiming to decrease the glucose affinity of *MAL11*, a difficulty arises by the fact that utilization of sucrose and maltose is also dependent on the hexokinase reaction. In such cases, the deletion of hexokinase genes cannot be used to disable glucose conversion and instead the use of non-metabolizable glucose analogs (for instance, 2-deoxy-glucose [53]) in cultures could provide the desired counter-selective pressure on glucose transport by its intracellular accumulation.

Third, the driving force of the transport process mediated by the targeted transporter should be sufficient to achieve (toxic) levels of accumulation of the counter-selectable substrate. This can be especially challenging if the proton-coupled transport process does not result in net transfer of charge due to transport of an anion, as is for example the case for proton-coupled monocarboxylic-acid transporters such as Jen1 [54]. Proton symport of the negatively-charged lactate, acetate, pyruvate and propionate anions via this transporter is electroneutral, therefore only the chemical component of the proton-motive force contributes to its driving force. To promote toxic accumulation of lactate without the contribution of the electrical component of the proton-motive force, the evolution could be performed at a low extracellular pH to maximize the chemical potential difference of protons across the plasma membrane [55]. An evolution experiment aimed at losing the ability of Jen1 to transport lactate could then be performed in a strain devoid of all other lactic acid importers [56,57] and lactate dehydrogenase activity [58].

## Figures and Tables

**Figure 1 jof-08-00358-f001:**
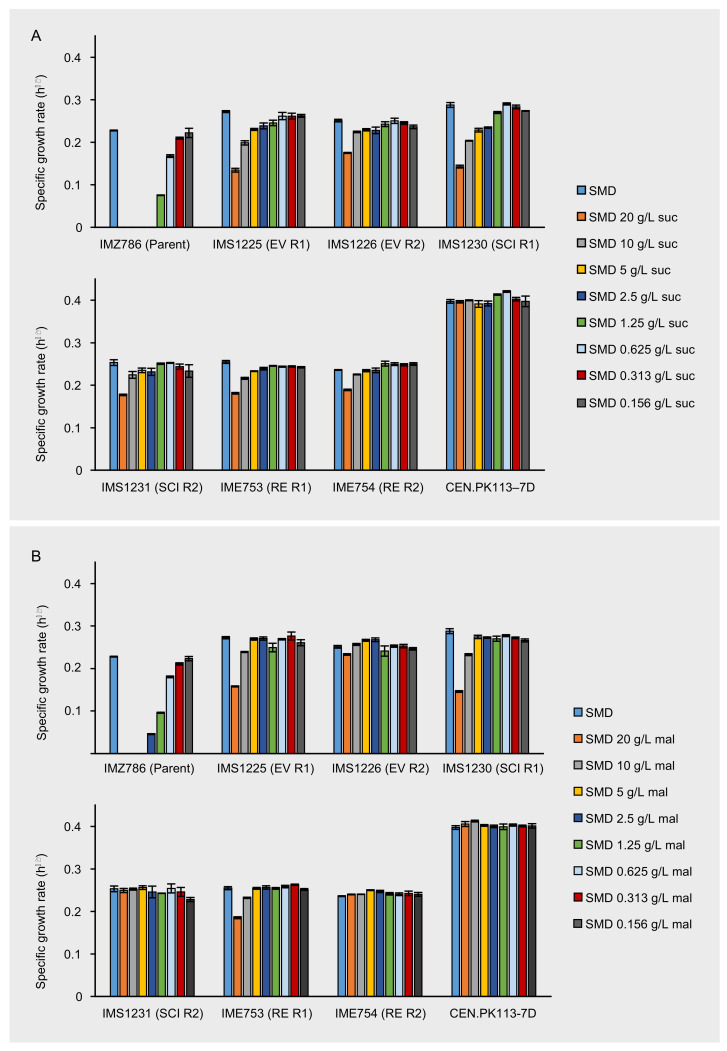
Maximum specific growth rates of IMZ786 (*sugar^0^ MAL11*) and strains that originate thereof, which were the evolved populations (EV) after laboratory evolution in glucose-limited continuous cultures with increasing addition of sucrose in two independent reactors (R1 and R2), the corresponding single-colony isolates (SCI) and the corresponding reverse engineered strains (RE). (**A**) Maximum specific growth rates on SMD within the presence of different concentrations of sucrose (suc). (**B**) Maximum specific growth rates on SMD with in the presence of different concentrations of maltose (mal). Data represent average and mean deviation of three replicate experiments. To validate the experimental procedure and data analysis, the growth rates of laboratory strain CEN.PK113-7D were also determined in all conditions.

**Figure 2 jof-08-00358-f002:**
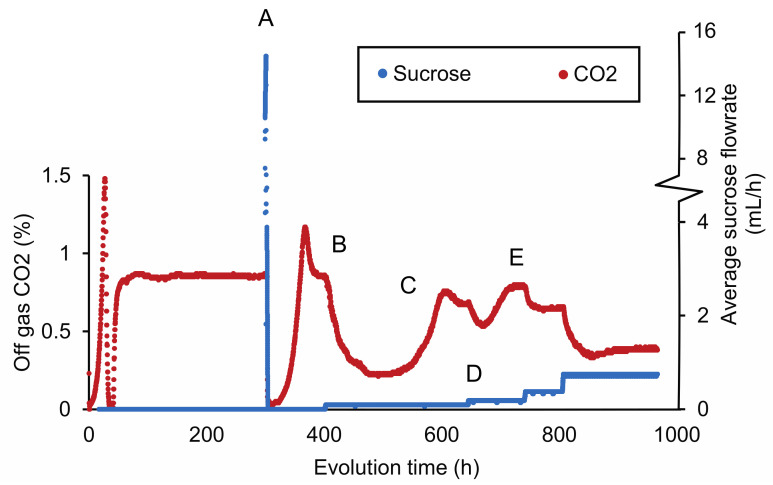
CO_2_ concentrations in the reactor off-gas and average flowrate of a 500 g L^−1^ sucrose solution during adaptive laboratory evolution of IMZ786 (*sugar^0^ MAL11*) in anaerobic glucose-limited chemostats at a dilution rate of 0.07 h^−1^ with pulse-wise addition of sucrose. The off-gas CO_2_ concentrations were used as a read-out to monitor growth and metabolic activity. One representative of the two independent evolution experiments is shown. (**A**) After a stable CO_2_ concentration of ~0.83% in the reactor off-gas was observed, sucrose was administered separately to the cultures at an average flowrate of 16 mL h^−1^. At this flowrate, the CO_2_ concentration decreased to 0.08% within 1 h. To subsequently prevent washout of cells and allow for evolution, the addition of sucrose was temporarily stopped. (**B**) After ~400 h, the average sucrose flowrate was set to 0.09 mL h^−1^. This resulted in a decrease of the off-gas CO_2_ concentration to a value of 0.22%. (**C**) The decrease in the off-gas CO_2_ concentration was, at the 480 h time point, followed by a strong increase, until it stabilized at ~0.64%. (**D**) Subsequently, the average flowrate of sucrose was increased to 0.19 mL h^−1^ and the culture was again left to evolve until a stable CO_2_ output was reached (**E**). An overview of the full experiment, which lasted ~3400 h, is presented in Appendix A.

**Figure 3 jof-08-00358-f003:**
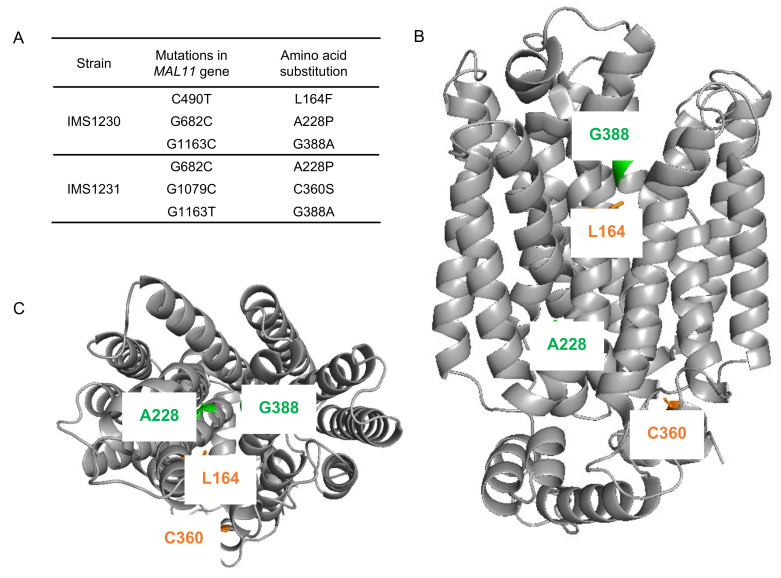
Mutations identified in *MAL11* after evolution of IMZ786 (*sugar^0^ MAL11*) in two independent, glucose-limited continuous cultures with increasing addition of sucrose. (**A**) In single-colony isolates from two independent reactors, three mutations were identified in *MAL11*. (**B**) (side view), (**C**) (top view). Structural model of Mal11, highlighting the location of the amino acid changes resulting from the mutations that occurred during evolution. Mutations that were identical in both strains are indicated in green, and mutations that were unique in each strain are indicated in orange. All structure figures were prepared with PyMol.

**Figure 4 jof-08-00358-f004:**
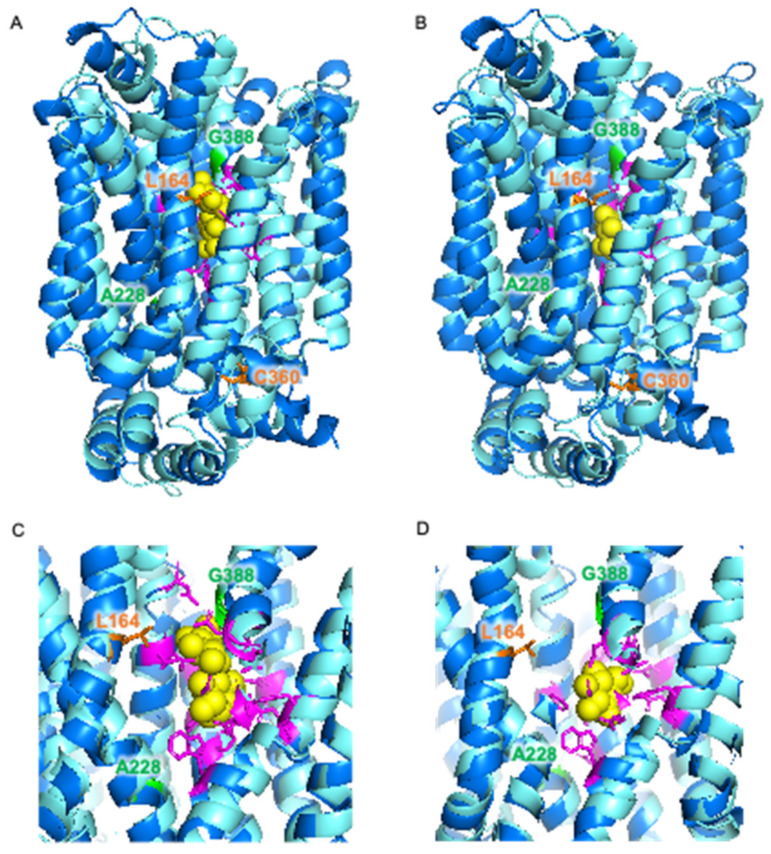
Superimposition of homology model of *S. cerevisiae* Mal11 (turquoise) [41] and crystal structures of human GLUT3 (blue) [42] bound to maltose (PDB: 4ZWB) (**A**,**C**) and bound to glucose (PDB: 4ZW9) (**B**,**D**). Glucose and maltose are represented by yellow spheres. Amino acid residues involved in sugar binding in GLUT3 are shown in magenta, and amino acid residues in Mal11 that were substituted during evolution of IMZ786 are shown in orange and green.

**Table 1 jof-08-00358-t001:** *S. cerevisiae* strains used in this study. The prefix ‘pUDE’ signifies an episomal plasmid.

Strain	Description or Relevant Genotype	Source
CEN.PK113-7D	*MATa URA3 HIS3 LEU2 TRP1 MAL2-8c SUC2*	[32]
IMK1010	*MATa ura3-52 LEU2 HIS3 MAL2-8C mal11-mal12::loxP mal21-mal22::loxP mal31-mal32::loxP mph2/3::loxP mph2/3::loxP-hphNT1-loxP suc2::loxP ima1Δ ima2Δ ima3Δ ima4Δ ima5Δ can1* *Δ::cas9-natNT2 hxt8* *Δ hxt14* *Δ gal2* *Δ hxt4* *Δ hxt1* *Δ hxt5* *Δ hxt3* *Δ hxt6* *Δ hxt7* *Δ hxt13* *Δ hxt15* *Δ hxt16* *Δ hxt2* *Δ hxt10* *Δ hxt9* *Δ hxt11* *Δ hxt12* *Δ stl1Δ*	[35]
IMZ786	*MATa ura3-52 LEU2**HIS3**MAL2-8C mal11-mal12::loxP mal21-mal22::loxP mal31-mal32::loxP mph2/3::loxP mph2/3::loxP-hphNT1-loxP suc2::loxP ima1Δ ima2Δ ima3Δ ima4Δ ima5Δ can1**Δ::cas9-natNT2 hxt8**Δ hxt14**Δ gal2**Δ hxt4**Δ hxt1**Δ hxt5**Δ hxt3**Δ hxt6**Δ hxt7**Δ hxt13**Δ hxt15**Δ hxt16**Δ hxt2**Δ hxt10**Δ hxt9**Δ hxt11**Δ hxt12**Δ stl1Δ* pUDE432 (*URA3 MAL11*)	This study
IMS1225	IMZ786 evolved in chemostats on SMD with addition of sucrose, first reactor, evolved population	This study
IMS1226	IMZ786 evolved in chemostats on SMD with addition of sucrose, second reactor, evolved population	This study
IMS1230	IMZ786 evolved in chemostats on SMD with addition of sucrose, first reactor, single-colony isolate	This study
IMS1231	IMZ786 evolved in chemostats on SMD with addition of sucrose, second reactor, single-colony isolate	This study
IME753	*MATa ura3-52 LEU2**HIS3**MAL2-8C mal11-mal12::loxP mal21-mal22::loxP mal31-mal32::loxP mph2/3::loxP mph2/3::loxP-hphNT1-loxP suc2::loxP ima1Δ ima2Δ ima3Δ ima4Δ ima5Δ can1**Δ::cas9-natNT2 hxt8**Δ hxt14**Δ gal2**Δ hxt4**Δ hxt1**Δ hxt5**Δ hxt3**Δ hxt6**Δ hxt7**Δ hxt13**Δ hxt15**Δ hxt16**Δ hxt2**Δ hxt10**Δ hxt9**Δ hxt11**Δ hxt12**Δ stl1Δ* pUDE1222 (*URA3 MAL11^C490T/G682C/G1163C^*)	This study
IME754	*MATa ura3-52 LEU2**HIS3**MAL2-8C mal11-mal12::loxP mal21-mal22::loxP mal31-mal32::loxP mph2/3::loxP mph2/3::loxP-hphNT1-loxP suc2::loxP ima1Δ ima2Δ ima3Δ ima4Δ ima5Δ can1**Δ::cas9-natNT2 hxt8**Δ hxt14**Δ gal2**Δ hxt4**Δ hxt1**Δ hxt5**Δ hxt3**Δ hxt6**Δ hxt7**Δ hxt13**Δ hxt15**Δ hxt16**Δ hxt2**Δ hxt10**Δ hxt9**Δ hxt11**Δ hxt12**Δ stl1Δ* pUDE1223 (*URA3 MAL11^G682C/G1079C/G1163T^*)	This study

## Data Availability

Not applicable.

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
