# Peer review of "Novel Evolutionary Engineering Approach to Alter Substrate Specificity of Disaccharide Transporter Mal11 in Saccharomyces cerevisiae"

_jof, 2022, doi:10.3390/jof8040358_

Round 1

Reviewer 1 Report

In the manuscript entitled “Novel Evolutionary Engineering Approach to Alter Substrate Specificity of Disaccharide Transporter Mal11 in Saccharomyces cerevisiae” authors describe how they prepared yeast Mal11 transporter with modified enzymatic properties (changed selectivity of the transporter) by simulated evolution. They identified the specific mutations that affect the selectivity of the transporter. The experiments are well designed and results are reasonably interpreted. I did not find any significant flaws.

I believe that the results described in the manuscript will be of interest both to people in the field of membrane transport as well as to those in yeast community who may use the similar approach in other fields of yeast biology.

The manuscript is written in good english.

I only have one minor concern: 

Lines 176 - 190 - Should be left out (these are instructions to authors accidentally left in the manuscript).

Author Response

We thank the reviewer for the positive assessment of the work and for pointing out the author instructions in the materials and method section. We have removed Lines 176-190 from the manuscript accordingly.

Reviewer 2 Report

In the manuscript ‘Novel Evolutionary Engineering Approach to Alter Substrate Specificity of Disaccharide Transporter Mal11 in Saccharomyces cerevisiae’, the authors presented an evolutionary strategy in yeast to alter the substrate specificity of disaccharide transporter Mal11, which has affinity for glucose, sucrose and maltose. Evolution in glucose-limited conditions with pulsed sucrose solution resulted in spontaneous mutant yeasts less sensitive to sucrose and maltose. In two independent experiments, three mutations occurred in MAL11, responsible for the disaccharide-insensitive phenotype.
In my opinion, presented topic might be interesting to the other fungal researchers working with transporters. However, the authors showed preliminary observations and hypothesis only, without explaining molecular mechanism and overall theory, which might support results
Major remarks:
Lack of kinetics of mutated transporters. Michaelis and transportation constant (Km, K, Vmax) should help to elucidate mechanism of transportation through ‘catalytic’ pore and confirm hypothesis included in the title (‘...substrate specificity...’! – Km, where it is in the manuscript?).
Moreover, dissection of third, unrepeated mutation (L164F and C360S) should bring some information, how the protein chain compensate sterically altered substrate pore.
Phenotypic plate tests of yeast transformants would be informative for the readers
Minor remarks:
Row 52: adaptive laboratory evolution (ALE)
Row 114: LiAc
Row 176-190 ??? some copy-paste error
Row 432-444: in my opinion, this paragraph is out of scope
In my opinion the paper, although dealing with a very interesting topic as a short report, is lacking elucidation of mechanism, involved in presented observation, necessary to form complete publication for JoF.

Author Response

We thank the reviewer for a thourough assessment of the work and answer each of the reviewers comments point-by-point below.

Comment: Lack of kinetics of mutated transporters. Michaelis and transportation constant (Km, Kt, Vmax) should help to elucidate mechanism of transportation through ‘catalytic’ pore and confirm hypothesis included in the title (‘...substrate specificity...’! – Km, where it is in the manuscript?).

Answer: We agree with the reviewer that the addition of in vivo experiments on transport kinetics would be a very relevant addition to the work. As we mention in the discussion (Lines 357-364) we hypothesize that there can be multiple mechanistic explanations for the observed phenotypes. Unfortunately, we were not able to include in vivo determination in the present work, but we believe that our physiological observations are sufficient to make a strong case that the ‘substrate specificity’ of the transporter has changed. In each of the four hypotheses (decreased disaccharide binding, lower disaccharide transport rates, increased glucose binding, decreased proton binding) the observed phenotype is the result of a change in substrate specificity of the transporter.

Comment: Moreover, dissection of third, unrepeated mutation (L164F and C360S) should bring some information, how the protein chain compensate sterically altered substrate pore.

Answer: We agree with the reviewer that the L164F mutation is of interest and have expanded our hypothesis about this mutation in the discussion. The sentence “Here, substitution of the leucine side chain by the bulky aromatic side chain of phenylalanine could sterically hinder binding of disaccharides.” was added.

Comment: Phenotypic plate tests of yeast transformants would be informative for the readers

Answer: We contemplated the use of plate tests in our study as they are visually appealing and straightforward to interpret. However, we decided to use liquid media cultures (Figure 2) as it also allows for phenotypical characterization and differences in growth rates can be quantified.

Comment: Row 52: adaptive laboratory evolution (ALE)

Answer: We thank the reviewer for the suggestion and the text has been changed accordingly.

Comment: Row 114: LiAc

Answer: We thank the reviewer for pointing out the error and corrected the abbreviation.

Comment: Row 176-190 ??? some copy-paste error

Answer: We thank the reviewer for pointing out the author instructions in the materials and method section. We have removed Lines 176-190 from the manuscript accordingly.

Comment: Row 432-444: in my opinion, this paragraph is out of scope

Answer: We consider a description of factors that impact the net driving force of transport (and thereby the achievable accumulation level) to be of relevance. However, we think we might not have made it sufficiently clear that these concerns are not only specific for transport of carboxylic acids (which we agree would be out of scope), but any negatively charged molecule co-transported with a proton. To better indicate that these considerations are relevant for transport of anions, the discussion was slightly expanded: “This can be especially challenging if the proton-coupled transport process does not result in net transfer of charge due to transport of an anion, as is for example the case for proton-coupled monocarboxylic-acid transporters such as Jen1 [54].

Comment: In my opinion the paper, although dealing with a very interesting topic as a short report, is lacking elucidation of mechanism, involved in presented observation, necessary to form complete publication for JoF.

Answer: We agree with the reviewer that in vivo elucidation of the mechanism would be an excellent next step, which unfortunately could not be performed in the present work. However, we are convinced that the elaborate physiological characterization, where 8 strains were grown in 16 different conditions to investigate the effect of the transporter mutations on strain physiology already provides valuable insights into functioning of the altered transport protein and warrants consideration for publication.

Reviewer 3 Report

The study of Claire de Valk and Mans explore the role of amino acid mutations that can alter the substrate specificity of sugar transporter protein Mal11 in yeast. Instead of studying particular amino acid mutations, the authors follow the approach of adaptive lab evolution that enriches for mutations with a desired phenotype. They posit that uptake of sugar via proton symporter through proton-motive force can provide an evolutionary benefit in a strain devoid of all sugar transporters. Analysis of the evolved strains and NGS shows enrichment of amino acid mutations that can cause change in substrate specificity due to proximity from sugar binding pocket.

The extensive experiments carried out in the study merit publication and the manuscript is very well-written, especially the Introduction and Discussion sections. The Results are also well described with inferences drawn and speculation about possible causes. This makes the manuscript well-informed. There are a few minors errors that I list below which I hope the authors make corrections and the manuscript will be ready for acceptance.

  1. Line 96: Spelling error “…that where less sensitive…”
  2. Lines 176-190: I believe these are paragraphs for manuscript requirements. Please remove these from the manuscript text.
  3. Line 327: Grammatical error, “Superimposition of the Mal1 homology…”

Author Response

We thank the reviewer for the positive assessment of the work and pointing out the instructions in the materials and method section. We have removed Lines 176-190 from the manuscript. We have also corrected the errors in line 96 and 327.

Round 2

Reviewer 2 Report

The manuscript ‘Novel Evolutionary Engineering Approach to Alter Substrate Specificity of Disaccharide Transporter Mal11 in Saccharomyces cerevisiae’ corresponds to an updated version of a manuscript previously submitted to Journal of Fungi. Again, the authors have shown that a set of mutations in Mal11 transporter altered its substrate specificity.

In my opinion, the authors did not explain molecular mechanism of mutated transporter activities by in vitro experiments to show and compare their substrate specificity, mentioned in the title. Without this result, the presented study is unfinished in the presentation of Mal11 substrate specificity for the journal such as JoF.

Author Response

We agree with the reviewer that the addition of in vivo experiments on transport kinetics would be a very relevant addition to the work. As we mention in the discussion (Lines 357-364) we hypothesize that there can be multiple mechanistic explanations for the observed phenotypes. Unfortunately, we were not able to include in vivo determination in the present work, but we believe that our physiological observations are sufficient to make a strong case that the ‘substrate specificity’ of the transporter has changed.

In the physiological setup, transport of glucose by Mal11 is essential to provide building blocks and energy for growth. Furthermore, Mal11 transport activity is also essential to elicit the toxic effect of disaccharide addition, even in the absence of glucose (Lines 209-218). Therefore, while not a complete substitute for in vitro measurements, we believe that based on the physiological observations we can confidently state that the substrate specificity of the evolved Mal11 transporter has changed.

In the discussion we state four mechanistic hypotheses (decreased disaccharide binding, lower disaccharide transport rates, increased glucose binding, decreased proton binding) based on the observed phenotypes and each would be the result of a change in substrate specificity of the transporter.